# “Should I Inhale?”—Perceptions, Barriers, and Drivers for Medicinal Cannabis Use amongst Australian Women with Primary Dysmenorrhoea: A Qualitative Study

**DOI:** 10.3390/ijerph19031536

**Published:** 2022-01-29

**Authors:** Justin Sinclair, Susanne Armour, Jones Asafo Akowuah, Andrew Proudfoot, Mike Armour

**Affiliations:** 1NICM Health Research Institute, Western Sydney University, Penrith 2571, Australia; 20656687@student.westernsydney.edu.au (S.A.); 20355092@student.westernsydney.edu.au (A.P.); 2Agricultural Economics, Agribusiness and Extension, Kwame Nkrumah University of Science and Technology, Kumasi 0233, Ghana; asafojones60@gmail.com; 3Translational Health Research Institute, Western Sydney University, Penrith 2571, Australia; 4Medical Research Institute of New Zealand (MRINZ), Wellington 6012, New Zealand

**Keywords:** primary dysmenorrhea, period pain, cannabis, medicinal cannabis, regulations

## Abstract

Objective: This study sought to investigate the perceptions, barriers, and drivers associated with medicinal cannabis use among Australian women with primary dysmenorrhea. A qualitative study via virtual focus groups involving 26 women experiencing regular, moderate, or greater menstrual pain explored categories including cost, associated stigma, current drug driving laws, community and workplace ethics, and geographical isolation within the context of patient access under current Australian laws and regulations. Results: A qualitative descriptive analysis identified that dissatisfaction with current management strategies such as over-the-counter analgesic usage was the key driver for wanting to use medicinal cannabis. A number of significant barriers to use were identified including patient access to medical prescribers, medical practitioner bias, current drug driving laws, geographic location, and cost. Community and cultural factors such as the history of cannabis as an illicit drug and the resulting stigma, even when prescribed by a medical doctor, still existed and was of concern to our participants. Conclusion: Whilst medicinal cannabis is legal in all states and territories within Australia, several barriers to access exist that require government regulatory attention to assist in increasing patient adoption, including possible subsidisation of cost. The high cost of legal, medicinal cannabis was a key factor in women’s choice to use illicit cannabis. Overall, the concerns raised by our participants are consistent with the broader findings of a recent Australian Senate inquiry report into barriers to patient access to medicinal cannabis in Australia, suggesting many of the issues are systematic rather than disease-specific. Given the interest in use of medicinal cannabis amongst women with primary dysmenorrhea, clinical trials in this area are urgently needed.

## 1. Introduction

Primary dysmenorrhea (PD) is the most common cause of period pain (dysmenorrhea) in the absence of any underlying pelvic pathology in adolescents and young women [1]. Period pain is estimated to affect about three-quarters of all women worldwide under the age of 25 [2] and over 90% of young women under 25 in Australia [3,4]. Aside from painful menstrual cramps, other associated symptoms women commonly report include diarrhoea, pain radiating into the back and thighs, headaches, fatigue, fever, light-headedness, nausea, and vomiting [5,6]. This results in a variety of negative impacts, including absenteeism from school or tertiary education both in Australia [3,4,7,8] and worldwide [2], as well as significant productivity losses in the workplace [9].

Current consensus guidelines [10] and systematic reviews recommend non-steroidal anti-inflammatory drugs (NSAIDs) as first-line medications for most women experiencing PD [11], with hormonal therapies such as the combined oral contraceptive (COC) pill to be considered in women not currently planning pregnancy or those that experience NSAID adverse effects [12]. However, a significant number of young women report that analgesic usage does not adequately control their pain [13,14], leading to women commonly reporting a variety of non-pharmacological management strategies to manage their pain and other symptoms, with specific choices being influenced by menstrual health literacy, geography, and cultural background [13,15,16,17,18,19]. There is some evidence for effectiveness of some of these self-management strategies, including exercise and sustained heat [20,21], but not for others [22]. Given the strong correlation between the severity of menstrual pain and the negative impact [3], ensuring effective pain management options are available for young women is vital.

The use of cannabis for female reproductive complaints has a long ethnopharmacological history, but little modern clinical evidence in the literature. The earliest records associated with cannabis use therapeutically are associated with the Chinese, Indian, and Egyptian cultures before the Common Era [23,24,25], with the earliest Chinese pharmacopoeia, the *Shen-nung pen ts’ao ching*, ascribing cannabis use for rheumatic pain and disorders of the female reproductive system [24,25]. Historical medical evidence of cannabis being used to manage dysmenorrhea dates back to the 1800s, with Reynolds recommending its use in *The Lancet* in 1890 [26]. Previous research by our team has found that many women in Australia [27] and New Zealand [28] with endometriosis report using cannabis, mostly obtained illicitly, to manage their menstrual and pelvic pain, and there is a significant willingness of women in the USA to use cannabis for gynaecological conditions, particularly those associated with pain [29]. However, very little is known about the suitability and acceptability of using cannabis to manage primary dysmenorrhea. We know from our team’s previous research that significant pain reductions alone do not necessarily mean women will continue to use an intervention and that other factors such as cost are important in the choice of management strategies [30] and therefore before potentially undertaking clinical trials in this area we aimed to explore the perspectives of Australian women on the barriers and drivers of the potential use of cannabis for primary dysmenorrhea.

## 2. Materials and Methods

### 2.1. Population and Sampling

This study was approved by the Western Sydney University Human Research Ethics Committee (HREC) on the 4 November 2019 (Approval #H13538). Women were eligible to participate in the focus groups if they were currently living in Australia, were aged 18 years and over, and had experienced moderate or greater (4/10 or greater on a 0–10 numeric rating scale) period pain for at least two of their last three periods. Women were excluded if they had a diagnosis of secondary dysmenorrhea such as endometriosis or adenomyosis. Recruitment text specified this was a discussion of managing period pain symptoms and the potential role of medicinal cannabis. Previous usage of cannabis was not a requirement for participation in this research project. Recruitment was conducted via social media advertising on Facebook and Instagram, as well as social media postings by Family Planning New South Wales. Study participants who responded to calls of interest via social media were invited to participate via an anonymous online portal, with participant screening being conducted and written informed consent obtained prior to the focus groups. Focus groups were conducted via Zoom^®^, and all participants were asked to not use their real names whilst also keeping their cameras turned off to maintain anonymity. This was also important to ensure participants felt more comfortable discussing potentially illegal activity. Four online focus groups in total were run, comprised of Group 1: 18–25 years, Group 2: 26–35 years, Group 3: 36 years and over, and Group 4: mixed ages. This was due to the presupposition that the symptom severity and management strategies may potentially vary by age.

### 2.2. Data Collection

Self-care or self-management was defined as pharmacological, non-pharmacological, physical, or psychological strategies for managing PD symptoms [13]. In this study the term ‘medicinal cannabis’ refers to a quality-assured and standardised cannabis product that has been prescribed by a medical doctor, and ‘self-prescribed’, when used in the context of cannabis, refers to cannabis obtained through illicit supply chains but used for therapeutic purposes. Topics covered in the focus groups explored both barriers and drivers for cannabis usage, including shortcomings with current management strategies, cost, legality of driving and working, geographical restrictions, and the stigma still associated with cannabis use both in the community and workplace. Appendix A provides the questions used as starting points in the focus groups.

### 2.3. Data Analysis

A qualitative descriptive approach was used to analyse and code the data [31]. Qualitative descriptive research provides a summative account of an everyday experience, is presented in everyday language [32], and seeks to describe the meaning participants attribute to their experience as close to the fact as possible [33]. A qualitative content analysis was undertaken [34]. Initial meaning units were condensed into codes by one researcher (S.A.). Codes with similar patterns were grouped into categories and subcategories. This process and the subsequent categorisation were discussed with the first author (J.S.) and senior author (M.A.). After a consensus was reached regarding classification of codes and subsequent categorisation, quotations in each domain were summarised and presented in a qualitative descriptive manner. To ensure anonymity but to allow identification of different participants throughout various categories, we ensured that each quotation indicated which focus group was used (FG) and which participant in that group was speaking (P) (e.g., Participant 3 in Focus Group 1 would be represented as FG1, P3).

## 3. Results

Twenty-six women participated in the online focus groups. The median age of participants was 32 years old and ranged from 22 to 47 years old. The demographic information of participants is presented in Table 1. All women had associated menstrual symptoms in addition to the menstrual pain itself. Participants reported experiencing symptoms of mood changes (e.g., irritability, mood swings), fatigue and lethargy, dizziness, light headedness or feeling unbalanced, food cravings or loss of appetite, bloating, diarrhoea and/or constipation, back pain, leg pain, nausea, and breast tenderness. More severe associated symptoms self-reported were anxiety, fainting, insomnia, vomiting with pain, premenstrual dysphoric disorder (PMDD), and migraine-type headaches. To the best of our knowledge, all self-reported cannabis use by participants in this paper was self-prescribed.

### 3.1. Category 1: Not Many Options

This category captured that women with moderate or greater period pain struggled to find adequate relief with their current treatment plans, which mostly consisted of NSAIDs and heat. Three quarters of participants described previous experiences with the use of self-prescribed cannabis, but only about half of these were during menstrual cramps. Most respondents felt significant reduction in their menstrual pain when used; however, some reported that previous recreational usage had unpleasant side effects, and that different modes of administration may have different effects—both positive and negative. Despite these concerns the majority of participants, irrespective of past usage, indicated that due to lack of effective symptom management they would be open to trying medicinal cannabis for their menstrual symptoms.


*“…. And I found cannabis and melatonin were the only two things that would work and not get me a hangover the next day. It also helps with the pain as well because if you’re high, you’re just enjoying life and the pain goes away, you don’t really notice it at all. And it helped with the food craving as well”*
(FG3, P5)


*“I did smoke a bit of a joint at a party but it made me feel paranoid…most of my positive experiences were with vaping”*
(FG2, P6)

### 3.2. Category 2: You Don’t Always Have to Inhale

This category captured the fact that participants often wanted options in their cannabis usage, especially with respect to future medicinal cannabis usage. Whilst medicinal cannabis options in Australia include sublingual, oral oil and capsule dosage forms, self-prescribed illicit cannabis supply chains lack such sophistication and limit choices. Some current cannabis consumers inhaled cannabis, either using a vaporiser or in a rolled joint, in order to more easily control the dosage for fear of being too “high”. Other participants explained they would prefer taking cannabis either by capsules or oral oil due to being conscious of the effects of inhaling the vapour and to avoid the smell either in the house or on their clothes. Overall, the goal for most participants was to be able to reduce the pain but also avoid feeling too ‘high’, so that they could still perform their normal activities.


*“…I use the vape pen at the moment because I like to control the dose. I like that it’s fool proof. You fill it, you turn it on, you smoke it, you’re done.”*
(FG1, P6)


*“The oil and the capsules would be, by far, the most acceptable. The vapour, I probably wouldn’t be willing to try just because I’m a little bit unsure about the knowledge and evidence around long-term side effects or inhaling of anything that’s vapourised”*
(FG2, P4)

### 3.3. Category 3: Go Directly to Jail?

This category captured participants’ comments and awareness regarding driving after medicinal cannabis use, roadside drug testing, and legal ramifications. Australia, as the first country to introduce roadside drug testing laws [29], makes no distinction between medically prescribed or illicitly sourced cannabis when delta-9-tetrahydrocannabiniol (THC—the main psychotropic cannabinoid found in cannabis) is detected in point of collection bodily fluid testing. The uncertainty around the legality of driving after the use of medicinal cannabis, even when they are not feeling impaired, was a significant concern to most women, as almost all had to drive due to either work or family commitments such as dropping off and/or picking up children. Participants were especially concerned about the confusing, seemingly contradictory legal outcomes should they have a positive roadside drug test. While the women were clear about the legal alcohol limit and how to manage this, they were unsure about how to manage the issue of driving and how long after cannabis usage it would be acceptable to drive:


*“…What are the implications? Because there’s no laws yet about—you know that you can have a certain amount of alcohol and then you’re safe to drive, but there is none of those contingencies around cannabis and the current tests are basically what’s in your system, you’re done, and I think that’s a real limitation of our law.”*
(FG3, P2)

In addition to the legality of driving, some participants expressed concern that drug testing at work may detect their medicinal cannabis use and cause legal problems. Even if the prescription and consumption was legal, there was a lack of clarity around how this would interact with workplace health, safety, and employment contract laws that make no distinction between the presence of cannabis THC metabolites *per se* and the degree of impairment (if any) in the worker at the time of being tested.


*“A barrier for me would be a work issue. We actually get drug tested at work quite often, so that’s something I have to be aware of… The other thing is driving, what happens if we get pulled over and drug tested?”*
(FG3, P2)

### 3.4. Category 4: The Stigma Persists

This category captured the perceived damage to their professional or social standing if their use of medicinal cannabis became known, despite its legality. Due to the stigmatisation of cannabis over the last 90 years, participants spoke about using the drug in a clandestine manner in order to maintain the existing social ties with community members, despite its supposedly legal status if prescribed medicinally. These concerns were similar irrespective of previous use of cannabis or not:


*“With co-workers and your workplace, your employers, if they knew—I mean I don’t know how you would know unless you disclose it to them, but if they caught wind that you were using medicinal cannabis, I think their perception of you would probably be—I don’t know, potentially damaging.”*
(FG2, P3)


*“I’m actually not concerned about my work people. It’s more other people in the community organisations and community work that I do. There’s a lot of stigma in the community groups that I’m involved with around cannabis and it would be frowned upon and looked on quite negatively, especially someone of my responsibilities and how people perceive me as what I do in the community.”*
(FG2, P4)

### 3.5. Category 5: You’ve Got to Know the Right People

This category captured the difficulties related to the limited opportunities to obtain medicinal cannabis for regional dwellers. These were due to the sometimes vast distances involved in Australia between regional, rural, and urban areas. However, participants also indicated that some physicians became uncomfortable and considered their usage or desire for usage of cannabis for treatment as unusual, and they often did not have a large range of GPs to choose from, and in some cases, no medical practitioners in their area would prescribe.


*“Sometimes in rural community, we only have limited opportunity or option to shop around for a good GP. So you come out against any GP’s personal opinions of the treatment and then also their knowledge of the treatment. If it’s not something like they believe in, they’re not gonna have all the information or won’t be comfortable prescribing it or I might not be able to access it.”*
(FG3, P6)


*“I live regionally and I don’t know—if there’s no one in my town who is willing to go down that track, then I would have to travel quite far, possibly to a capital city to get access.”*
(FG3, P5)

### 3.6. Category 6: “Cost Is Definitely an Issue”

This category captured that while all participants recognised the high cost currently for legal products, there were differing views among respondents on the acceptability of cost for legal products (varying between AUD $10/day, $20/day, $250/pa, $400–500/pa.). While difficulty in finding a suitable health professional to prescribe medicinal cannabis was identified as a barrier in Category 5, the difference in cost between illicitly sourced and medicinal products was a major driving factor in the use of illicit products for those who currently used self—prescribed cannabis. Four participants had no issue with cost as long as it was within reason and the quality of the product was good; however, most participants felt that the cost needed to be at least comparable with their current over the counter products:


*“I’d probably be willing to pay more than to whatever it is, 12 bucks for my Naprogesic at the chemist. If it was gonna help and I knew I was taking a natural substance rather than a pharmaceutical substance.”*
(FG1, P5)


*“It would have to be pitched at something that was very simply affordable. So if you’re having one period a month, maybe I’d pay 20 bucks a month but I certainly don’t think I’d pay more than that.”*
(FG3, P5)

## 4. Discussion

Participants in our study felt their current treatment regimes, consisting mostly of NSAIDs, heat, and other non-pharmaceutical management, did not provide them adequate pain control, a common issue for women with PD [14], and this was the key driver for cannabis usage. Participants who had previously used cannabis reported it was often effective for pain symptoms but most participants, regardless of previous use, were concerned about the legal ramifications of driving or working (even if cannabis were to be medically prescribed); negative judgement from family, friends, and colleagues; and safety concerns over the various modes of administration of medicinal cannabis, especially those using inhalation, despite it being the most common method they used. They were also concerned about getting too “high”, and while this was not always unpleasant or unwanted, it was not the goal of their cannabis use. Furthermore, other unwanted side effects such as paranoia and the unknown long-term safety of inhaling cannabis [35] were raised as concerns, which are particularly poignant for those using illicit cannabis for therapeutic purposes (i.e., not medically prescribed) which is not standardized or undergone quality assurance testing. Participant responses exemplified the perception that while cannabis, if prescribed by a doctor, might be medicinal and even legal, it was somehow less “legitimate” than standard pharmaceutical drugs including opioids, and they were more likely to face repercussions, whether they be social and/or legal, for its use.

Due to the varying maturity of countries regulations and associated industry development (e.g., cultivation, manufacturing) of both medicinal and recreational cannabis, costs for medicinal cannabis products can vary greatly around the world. Australia implemented a medical cannabis scheme in 2016 via amendment to the *Narcotic Drugs Act* of 1967, and due to current industry immaturity, patients still rely heavily on imported products from countries such as Canada. Findings from the Australian Senate inquiry into current barriers to patient access to medicinal cannabis by the Community Affairs References Committee identified cost as one of the biggest barriers for patients struggling to access medicinal cannabis [36]. Costs associated with medicinal cannabis can be classified into two major categories: cost of the medicinal cannabis product itself and the cost for obtaining an approved prescription from a medical doctor. A recent industry report from 2020 [37] suggests that the average patient spend per month (dictated by product brand and dose prescribed) for a cannabis product is AUD $384 (approximately AUD $12.80 per/day), a reduction from AUD $436 per month (approximately AUD $14.53 per/ day) at the start of 2020. The potentially high cost of medicinal cannabis products are a significant barrier, and when asked what costs would be considered acceptable for such products, overall, an AUD $12–20 per month price range was considered reasonable (e.g., equivalent to monthly expenditure for NSAIDs etc.) and participants indicated a willingness to pay for effective treatment. Additional burdens to cost of medicinal cannabis are associated with the prescription obtained from the medical doctor, often through specialist cannabis clinics. A submission from the Australian Pain Management Association to the Senate Inquiry [36] states patients describe costs of between AUD $300 and $500 for initial appointments at such clinics, many of which are done through telehealth. Such a fee would be in addition to the product cost. Our participants indicated that given the current Australian cost (January 2021) for NSAIDs available over the counter, such as ibuprofen (AUD $5.50 for 24 caplets) or Naprogesic (AUD $11.25 for 24 tablets (naproxen sodium 275 mg)), these products were relatively affordable and likely to last several menstrual cycles, whereas current medicinal cannabis pricing is prohibitively expensive. Therefore, given the current out of pocket cost for medicinal cannabis, respondents indicated they were likely to either stay utilising current pharmaceuticals, or potentially procure illicit sources of cannabis, as they felt these were a significantly cheaper option.

The fact that in Australia cannabis exists both as a legal medicine and an illicit drug raises another significant barrier to access, particularly associated with Australian drug driving laws and workplace drug testing. This has attracted a great deal of media attention, with a New South Wales magistrate recently resigning over laws he deemed as being both “grossly unfair” [38] and “destructive and ineffective” [39]. In what has been described as a “zero tolerance” approach to drug driving laws [40], mobile drug testing across all Australian states and territories criminalises the detectable presence of THC in oral fluids, urine, or blood, regardless of impairment. In essence, the mere presence of THC constitutes an offence. Penalties associated with this strict liability offence include a criminal record, fines, and driving licence disqualification, the latter of which is a seriously limiting factor to those that live in rural or remote communities. Aside from the Australian Senate inquiry report recommending a review of state and territory criminal legislation in relation to current drug driving laws and the implication for legal medicinal cannabis patients [36], a South Australian Magistrate dismissed charges against a legal medicinal cannabis patient for drug driving due to lack of evidence of impairment in what is one of the first legal test cases in Australia [41]. The fundamental basis of the drug-driving issue is the need to update the regulatory and legal frameworks to ensure that adequate road safety continues whilst allowing for legally prescribed cannabis patients to use their medicine unimpeded and without unjust disadvantage. Australian workers in high-risk industries, such as construction, defence, maritime and mining operations, and road and rail transport, are all similarly disadvantaged in a workplace relations context as they are also the occupations usually subjected to workplace drug testing policies [42].

Despite the increasing prevalence and acceptance of both medicinal and recreational cannabis worldwide, a pervading stigma is still associated with its use across communities, cultures, professional work environments, and family units, as was reflected by our participants. Stigma may be used as a form of social control and as a method to punish, marginalise, or discourage certain behaviours [43]. Previous qualitative research investigating the perceptions of cannabis as a stigmatised medicine [44] has demonstrated that various strategies employed by people to manage the stigma of using cannabis for therapeutic purposes included using cannabis responsibly, keeping use hidden or secret, and spending time educating those that do not understand cannabis use as a medicine, or who did not approve. The potential reputational damage or perception of not being responsible could be seen as a strong enough barrier to patient use within our cohort. Such stigma could also be applied to medical professionals, who not only have preconceived biases that are based on their previous education relating to the harms associated with cannabis use, but who are concerned about how fellow medical colleagues or influential colleges may view them if they start prescribing cannabis for their patients. Whilst all Australian medical practitioners are legally able to prescribe medicinal cannabis through the Therapeutic Goods Administration’s Special Access Scheme Category B application process, many are reticent to do so due to having little training in, or understanding of, the endocannabinoid system (ECS), cannabis pharmacology, or current evidence of therapeutic effect. Health professionals themselves note this as a key barrier to access in both prescribing medicinal cannabis and discussing it with their patients [36]. A lack of medical education around medicinal cannabis was a sub-category also identified by the Australia Senate inquiry report [36], which has provided several recommendations to remedy this situation. Despite the report findings on these issues now being over 12 months old, it is apparent that no movement toward enhanced training protocols has come about from the university medical faculties, specialist training colleges, nor the Australian Medical Council, despite the bipartisan senate committee calling on them all to do so.

Australia is a vast country of approximately 7.6 million square kilometres, with 29% (around 7 million people) of the population classified as living in rural or remote locations [45]. Australian citizens living in rural and remote regions have higher levels of disease and injury; live shorter lives; and have poorer access to, and use of, health services compared to those in metropolitan areas [46]. Additionally, a 2018 survey of 640 Australian general practitioners (GPs) noted that GPs generally rated their knowledge of medicinal cannabis as poor, with less than 10% understanding the current regulations pertaining to medicinal cannabis access [47]. As such, specialist cannabis clinics are emerging in major metropolitan capital cities such as Sydney, Melbourne, and Brisbane, with focus group participants suggesting that limited numbers of rural doctors, coupled with any preconceived biases around cannabis as a medicine, means they would need to consider travel to major cities to find the clinical assistance they require, which directly increases patient costs and burden on an already underserved population.

## 5. Limitations

We were unable to recruit women under the age of 22 to participate in our study, and hence these results may not reflect the views of the 18–22-year age group. Our analysis did not identify any trend in responses according to age group; however, our findings may not reflect the same drivers or barriers in this younger age group. Just under three quarters of our participants reported having previously used cannabis. This is significantly higher than usage in the general Australian population of 11.6% [48], and therefore it is possible that our participants have a different view on cannabis than the wider population. Given that generalisability to wider populations is not the necessary intended goal in this type of analysis [49], caution must be taken with extrapolating findings to the large number of women in Australia with PD.

## 6. Conclusions

Women with moderate or greater symptoms of primary dysmenorrhea reported shortcomings with their current self-management strategies and were positively inclined to try medicinal cannabis for pain and symptom management. Barriers to usage in this population are similar to those previously identified by a recent Australian Senate inquiry and include high cost and difficulty in finding medical practitioners who can or would agree to dispense medicinal cannabis, especially for those women living in rural or remote areas. The lack of clarity around driving laws and impairment in the workplace, especially in contrast to other substances such as alcohol, left women feeling unsure about how and when they could use medicinal cannabis. Given the high prevalence of primary dysmenorrhea and its wide-ranging negative impacts, clinical trials in this area are needed to determine what, if any, role medicinal cannabis may have in managing menstrual pain.

## Figures and Tables

**Table 1 ijerph-19-01536-t001:** Demographics of participants.

Characteristic	N	%
Location		
Urban area	20 (77%)	
Rural or regional area	6 (23%)	
Education		
Did not finish HS	1	3.8%
TAFE	5	19.2%
Bachelor’s degree	15	57.7%
Master’s degree	4	15.4%
Doctoral degree (e.g., PhD)	1	3.8%
Employment		
Full-time employee	15	57.7%
Part-time employee	6	23.1%
Self-employed	2	7.7%
Not currently employed	3	11.5%
Currently studying (tertiary education)	9	34.6%
Previous cannabis use		
Yes	19	73.1%
No	6	23.1%
Prefer not to answer	1	3.8%

## Data Availability

The data presented in this study are available on reasonable request from the corresponding author. The data are not publicly available due to disclosure of illegal activity.

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
