# Peer review of "“Should I Inhale?”—Perceptions, Barriers, and Drivers for Medicinal Cannabis Use amongst Australian Women with Primary Dysmenorrhoea: A Qualitative Study"

_ijerph, 2022, doi:10.3390/ijerph19031536_

Round 1

Reviewer 1 Report

The authors of this article succed in presenting and discussing the use of medicinal cannabis amongst Australian women in order to treat dysmenorrhoea. The health disadvantages of cannabis used should be more discussed since both the participants and the published articles refer to them. Moreover, since this is a polulation based article, data will be better presented in graphs and statistical analysis may help more conclusions to be conducted.

Author Response

Thank you for your comments. Please see attached our reviewer response.

Kind regards,

Justin Sinclair (on behalf of all authors)

Reviewer 2 Report

Thank you for the opportunity to review the manuscript titled "Should I inhale?" - Perceptions, barriers, and drivers for medicinal cannabis use amongst Australian women with primary dysmenorrhoea: A qualitative study. 

First of all, congratulations on a very interesting and needed research. I have some recommendations for the manuscript. These recommendations are mainly centered around the distinction between cannabis use (as in recreational or self-prescribed use) and medicinal cannabis use (as in its prescribed use), a seemingly minor but important distinction.

Introduction
No comments

Materials and Methods
1. When describing participant recruitment, the authors explain that previous usage of cannabis was not a requirement for participation in the project. But, did the invitation mention the project was exploring the use of medicinal cannabis? This could help explain the large number of participants who used cannabis (mentioned as a limitation).

2. In the data collection section, the authors describe the topics covered during the focus groups (also included in Supplementary file S1). Here, they also explain the term "self-care" and "self-management." While people can use cannabis for therapeutic purposes, and they often do, medicinal cannabis is a term usually reserved for a health professional's prescribed use of cannabis. Could the authors help clarify this distinction and whether discussions focused on cannabis use as a self-care strategy and/or the prescribed use of cannabis?

Results
3. Line 137
The authors explain that most participants had previous experience using cannabis, but only half of these participants used it during menstrual cramps. Was this use self-prescribed?

4. Line 150
The authors explain that category 2, 'you don't always have to inhale', is a category that "captured that participants often wanted options in their cannabis usage." Was this part of a conversation about the potential use of medicinal/prescribed cannabis? The authors may consider adding some information about the types and forms of medicinal cannabis available in Australia.

5. Category 3 "go directly to jail?"
This category also exemplifies the importance of distinguishing between self-prescribed cannabis and medicinal cannabis. For example, the paragraph starting in line 180 reads:
"In addition to the legality of driving, some participants expressed concern that drug testing at work may detect their cannabis use and cause legal problems. Even if the prescription and consumption was legal, there was a lack of clarity around how this would interact with workplace health…"
It seems that participants expressed concerns about the risks of using cannabis as a self-management strategy and some confusion about whether these risks would remain when using medicinal (prescribed) cannabis, but this is not entirely clear.

6. Category 4
"This category captured the perceived damage to their professional or social standing if their use of cannabis became known, regardless of its medical legality or therapeutic effectiveness." 
- Were there participants who were using prescribed cannabis? (see also point 8, below)
- Did participants who didn't use cannabis also express these concerns about the stigma associated with cannabis use when discussing the potential use of medicinal cannabis? 
It is undeniable that the stigma surrounding cannabis will represent a barrier to its medicinal use (as further evidenced by this study), but clarifying these points could further emphasize this issue.

Discussion
7. Line 238 "and this was the key driver for cannabis usage." This statement doesn't seem to be supported by the presented data. As mentioned in the Results section, Category 1, not all participants used cannabis, and only half of those who used it had used it during menstrual cramps. 

8. Line 241. 
"(even if cannabis was medically prescribed)." If no participant had been prescribed cannabis, I would suggest rephrasing this statement to "even if cannabis were to be medically prescribed).

9. Line 244
"They were also concerned about getting too 'high'… it was not the goal of their cannabis use". This statement also doesn't seem to be supported by the provided data. As mentioned above, only half of the participants who used cannabis used it during menstrual cramps, and there are no further data regarding the goal of their cannabis use.

10. Line 278
"Or potentially procure illicit sources of cannabis, as they felt these were a significantly cheaper option." This information is not presented in the results section, and it is introduced to the reader for the first time in the discussion section.

Conclusion
11. Line 358.
"Were positively inclined to try medicinal cannabis for pain and symptom management." I suggest adding this statement to Category 1 in the results section. This category states that there are few options; about half of those who used cannabis used it during menstrual cramps; most of these participants felt a significant reduction in their menstrual pain when used. If, as a group, participants were inclined to use medicinal cannabis for pain and symptom management, the authors should also mention it in the results. 

Author Response

(The authors gave the same response as above.)

Reviewer 3 Report

The manuscript is clear and well presented and it represents a good starting point for future clinical trials to investigate the efficacy and safety of medicinal cannabis use to treat PD. The size of the focus group represents a limitation of the study but, at the same time, it's proof that the population is still reluctant to use cannabis for medical purposes, which is perceived as a drug. That supports the conclusion of the study, that more clinical evidence is needed to increase the trust in this type of medication and more regulations are necessary to clarify the legal issues regarding the use of cannabis as medicine. So, the research has achieved its goal and the results are important. 

Author Response

(The authors gave the same response as above.)

Reviewer 4 Report

This is a well-written manuscript about a relevant subject. I hope the authors consider the possibility of submitting an abridged version of this to a national medical or nursing journal. I have only a few comments.

  1. The Discussion could be shortened, and I lost the thread when reading the list of so many amounts for cannabis.

  1. R 71-73, "We know from our team’s previous research that significant pain reductions alone do not necessarily mean women will continue to use an intervention": I did not understand this part of the sentence.

  1. R. 264-266, "Cost was a significant barrier to most participants in our study, however, overall an AUD$12-20 per month price range was considered reasonable": I did neither understand this sentence. Cost was a barrier and yet they found the amount reasonable?

Author Response

(The authors gave the same response as above.)

Round 2

Reviewer 2 Report

Thank you for addressing the comments. I am satisfied with the included revisions.